# GENERATING OUTPUT DIVERSITY FROM PROMPT RE-TOKENIZATION

## ABSTRACT

Large language models (LLMs) process tokens, not characters, and in principle do not have access to subtoken structure. This is often seen as a fundamental flaw of tokenization, and has been pointed out as being responsible for many problematic behaviors on character-level tasks. However, we present evidence that trained LLMs do in fact learn subtoken structure, and that it can be leveraged in a novel sampling strategy which we call *tokenization sampling*, whereby different token encoding of the same surface string can lead to different completions. Through experiments on benchmarks in knowledge retrieval, mathematical reasoning, and code generation, we observe that while such retokenization tends to make easy problems slightly harder, it also allows LLMs to solve otherwise impossible problems, which conventional temperature sampling would *never* get correct. Thus, the redundancy of tokenizations of a given string offers a method of test-time data augmentation that can generate multiple-views of the same prompt. In sum, our work presents an unsung advantage of tokenized language modeling, which is a mechanism to generate a diversity of outputs from prompts which are identical at the byte level.

## 1 INTRODUCTION

Tokenization is a fundamental pre-processing step in language modeling that remains understudied. The creation of tokens typically relies on simple frequency counts of bigrams, and while tokens often capture meaningful structure within text, sometimes natural morphological structures end up getting tokenized in unnatural ways. Furthermore, large language models (LLMs) often utilize a fixed tokenizer, which potentially constrains their ability to learn the compositionality of tokens. For instance, if in a training run "probable" is always represented as a single token `probable`, it seems intuitively unlikely that a model will learn that the two-token sequence `prob` `able` represents the same word. This blindness to subtoken structure has been a source of embarrassment in the past for frontier models which could perform complicated mathematical and coding tasks, but seemingly could not count the letters in a word (Cosma et al., 2025).

Despite these shortcomings, we suggest that tokenization offers an advantage that has seemingly gone unnoticed. Tokenization introduces a gauge freedom in the encoding of a byte-string; there exist a huge number of token sequences that decode to the same surface string. While an LLM will typically only see a "canonical" token encoding, we find that LLMs appear to understand different tokenizations as being similar. This emergent symmetry under retokenization, combined with the *strict semantic equivalence* of different tokenizations, leads us to propose retokenization of a prompt as a form of test-time data augmentation (Dang et al., 2025; Chai et al., 2026). We explore how reto-kenization affects model performance on benchmarks in different domains. In our experiments, we show that sampling from distinct tokenizations produces a diversity of outputs in LLMs, which can in some cases help it access solutions otherwise impossible to reach using conventional sampling.

**Notation and Nomenclature**   Before proceeding, we want to set notation and nomenclature, making sure to be precise about terms that are often used in casual or idiomatic ways.

Raw text data is represented as a byte sequence, or byte-string. **Tokens** are multiple-byte sequences, typically learned with an unsupervised algorithm like byte-pair encoding (BPE) (Gage, 1994; Sen-

nrich et al., 2016), that might represent either meaningful combinations of character-level bytes (e.g. words) or frequently occurring subword strings like "ing". Formally, a **token vocabulary** of size $V$ is a bijective map from $[V]$, the set of natural numbers up to $V$, to a set of $V$ byte-strings $\mathcal{S}_V$. In this paper, we are not concerned with the learning of tokens, and take the token vocabulary to be fixed.

By **tokenization** we mean a map from a byte-string to a sequence of tokens, *given a fixed token vocabulary*. We denote this by the **token encoding** map $E : \mathcal{S} \to \mathcal{T}$, where $\mathcal{S}$ is the set of byte-level character strings $s$, and $\mathcal{T} = [V]^*$, the set of all sequences of integers from $[V]$. A given byte-string $s$ will have some number of possible token encodings, which we denote $L(s)$ (we give some examples in the appendix). A tokenization from this set will be denoted $E^\mu(s) = (E_1^\mu(s), E_2^\mu(s), ..., E_{T_\mu}^\mu(s)) \in \mathcal{T}$, where $\mu \in \{0, ..., L(s) - 1\}$. Importantly, the token encoding map $E$ is one-to-many, and therefore not a function. A **tokenizer** is a particular choice of token encoding $E^\mu$ for every $s$. This is typically done by using a ranked merge list from the BPE training to also encode a byte-string. A deterministic tokenizer is often employed to encode raw text data as token sequences; we refer to this as the **canonical tokenizer** $E^0(s)$, whereas $E^\mu(s)$ for $\mu > 0$ are non-canonical **retokenizations**. Fig 1A shows an example of a canonically tokenized word 'retokenize', along with some alternative tokenizations, which all utilize tokens from the token vocabulary.

**Emergence of Tokenization Symmetry**    If the byte-string is the fundamental object in language models, then tokens should not have any fundamental meaning, and tokenization must be a gauge freedom (Posfai et al., 2025). A gauge freedom or symmetry is a redundancy in the description of a system. A simple example from physics is the choice of coordinates, which can be transformed without affecting the dynamical laws. Examples of manifestly gauge invariant language models are byte-level models (Minixhofer et al., 2025; Hwang et al., 2025), process text explicitly as byte-strings. In fact, a tokenized language model which is required to be explicitly invariant to tokenization is equivalent to a byte-level model (see Appendix C).

However, in conventional tokenized language modeling, tokens are the fundamental objects. For this reason, tokenization is no longer a gauge symmetry. Indeed, since LLMs typically use a deterministic tokenizer during training, it seems unlikely that the decomposition of a word into all its possible tokenizations can be learned. Nevertheless, like Zheng et al. (2025), we find evidence that tokenization emerges as a conventional discrete symmetry in trained LLMs.

While there are regularization techniques which utilize stochastic tokenization (Provilkov et al., 2019), it is unclear to us at the time of writing whether the OLMo 2 family of models implements this type of regularization (OLMo et al., 2024). If not, then the pretraining and instruction fine-tuning dataset themselves must allow token substructure to be *seen*, and thus learned. What it is about this data that makes tokenization symmetry emerge remains a fascinating question for future work.

**Related Work**    Many works have pointed out unique challenges of tokenization, often to do with difficulties in character-level tasks (Singh & Strouse, 2024; Chai et al., 2024; Wang et al., 2024) and robustness to misspellings (Sperduti & Moreo, 2025). There are also many proposed regularization techniques, including fine-tuning with synthetically noisy data (Sperduti & Moreo, 2025; Feng et al., 2020) or pretraining with a stochastic tokenizer (Provilkov et al., 2019). Some works argue that even without such regularization, subtoken understanding actually emerges in larger models (Zheng et al., 2025; Cosma et al., 2025).

Prompt engineering (Liu et al., 2023) and test-time data augmentation (Bsharat & Shen, 2025; Chai et al., 2026) are powerful approaches to augmenting LLM performance at inference time. However, lack of robustness to prompt perturbation (such as synonym, oronym, or paraphrase in Qiang et al. (2024)) has been noted as a serious shortcoming.

## 2   TOKENIZATION SAMPLING

In this paper, we introduce tokenization sampling as an alternative to probabilistic temperature sampling for LLM generation. For a given prompt $s$, we generate a set of tokenizations $\{E^\mu(s)\}$, which includes the canonical tokenization $E^0(s)$. Starting with a retokenized prompt, we perform greedy

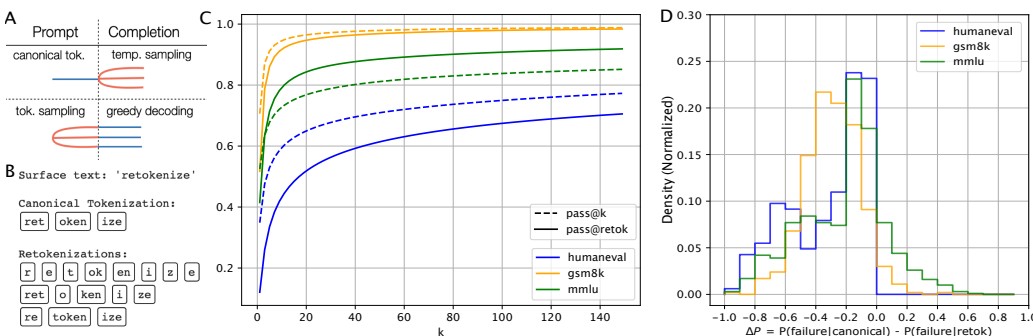

Figure 1: **A)** Illustrates the idea of tokenization sampling (bottom) versus conventional sampling (top): Canonical tokenization deterministically encodes a prompt, and stochastically samples completions. Tokenization sampling stochastically encodes a prompt, and deterministically generates completions. **B)** Shows an example of retokenizations. Every surface text has multiple encodings into token sequences, using a fixed token vocabulary. The canonical tokenization, produced by the model's tokenizer, represents only one of these. Sampling from a stochastic tokenizer has potential to produce more meaningful encodings. **C)** We observe similar behavior for pass@retok and pass@k, suggesting that retokenization samples meaningfully different perspectives of the same prompt. **D)** The distribution over three three datasets of the change in the failure probability between canonical sampling and tokenization sampling. Of particular note is the addition of mass above zero, indicating problems which benefit from retokenization.

sampling to generate an output. We introduce **pass@retok**(k), which is defined as the probability of generating the correct solution to a problem after sampling $k$ random tokenizations. This is analogous to the commonly reported pass@k, which is the probability of generating the correct solution after finite-temperature sampling of $k$ completions from a model starting with a canonically tokenized prompt. The difference is illustrated schematically in Fig.1A.

## 3 EXPERIMENTS

Fig 1C shows the pass@retok vs pass@k for benchmarks in three distinct domains: knowledge retrieval (MMLU), mathematical reasoning (GSM8k), and code generation (HumanEval). Overall, our results give strong evidence that tokenization symmetry emerges in the particular LLM we study (OLMo-2-1124-7B-Instruct). In the Fig. 2, we compare OLMo-2-7B models at different stages of training (base→ SFT → DPO → RLVR), and argue that this capability mainly emerges in post-training (OLMo et al., 2024).

**Knowledge Retrieval:** We explore the model's ability to perform knowledge retrieval and language understanding by testing on the Massive Multitask Language Understanding (MMLU) benchmark introduced by (Hendrycks et al., 2021). Fig. 1C shows that pass@retok actually rises faster than pass@k, although it has a lower asymptotic value. Fig.1D shows that for MMLU, the failure probability decreases for a significant fraction of problems under tokenization sampling. However, this benchmark is multiple choice, so one might worry that we are not giving a rigorous assessment of a model's understanding, since random chance would yield a curve which rises even faster.

**Mathematical reasoning:** A more challenging benchmark which cannot be solved by random guessing is the math word-problem dataset on grade school math (GSM8k) introduced by (Cobbe et al., 2021). Fig. 1C compares pass@k to pass@retok for a subset of 1000 questions on GSM8K. Although the values are starkly different for $k = 1$, tokenization sampling quickly becomes competitive with conventional sampling. Fig.1D shows a number (smaller than MMLU) but still non-negligible subset of problems for which retokenization helps reduce failure probability.

**Code completion:** Finally, we test our sampling on the code completion HumanEval benchmark introduced by (Chen et al., 2021). Similarly in this case, getting a correct solution is highly non-trivial, and unlikely to be the result of random chance. With this benchmark, only a small subset

of problems benefit from retokenization, as seen in Fig.1D. On this subset of problems, only re-tokenized prompts generate passing solutions. By iteratively merging tokens in these retokenized prompts, we can locally sample the space of tokenizations and find more prompts which generate passing solutions. For problems with many initial retokenizations that generated passing solutions (HumanEval problems 26 and 159), we do this local sampling for each initial retokenization. For more details, see Appendix A.5.1.

**Emergence of Tokenization Symmetry during Post-training**

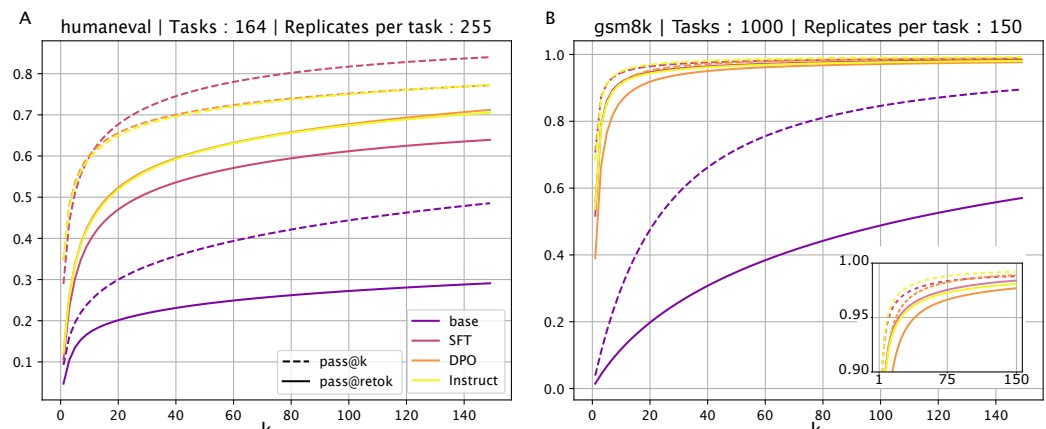

Figure 2: pass@k and pass@retok for the Base, SFT, DPO and Instruct variants of OLMo-2 7B parameter model family on **A)** humaneval, and **B)** GSM8k benchmark datasets. (Inset) Magnified view of the asymptotic region to highlight differences at high accuracy.

We extend our experiments to models at different training stages from the OLMo-2 7B parameter model family by obtaining pass@k and pass@retok for the GSM8k and HumanEval benchmarks for each model. These models belong to a sequence of training stages which include, in the order in which they are implemented: base pre-training, supervised fine-tuning (SFT), direct policy opti-mization (DPO), and reinforcement learning with verified rewards (Instruct). Across these datasets, both pass@k and pass@retok exhibit a sharp increase in models with post-training. While the evi-dence and mechanism of improvement in pass@k scores from post-training has been shown before (Chen et al., 2021; Cobbe et al., 2021; Ouyang et al., 2022), the reason behind a proportionate in-crease in pass@retok scores remains unclear. These results indicate a dominant role of post-training in the emergence of tokenization invariance in language models.

## 4 DISCUSSION

Tokenization can be annoying. It has been implicated as the root cause of embarrassing failures on elementary tasks (e.g. Cosma et al. (2025)), and has sometimes been referred to as a "curse" Chai et al. (2024) or "necessary evil" Vizuara. We offer a counterpoint to this sentiment, and argue that tokenization can be a boon to LLMs, offering a way to generate diverse outputs from prompts which are lexically and semantically identical. We have introduced the pass@retok metric to compare with the conventional pass@k, and found that sampling from tokenizations is comparable to conventional sampling. Furthermore, we see that the multiple views that retokenization provides allows the LLM to solve certain problems which seem impossible otherwise. These observations suggest that LLMs have an emergent symmetry with respect to retokenization of the prompt, one which must emerge during training. This work raises many questions which we hope to address in future work: how does this tokenization symmetry arise during the training process, and what structure of the data will give rise to it? How does the diversity of outputs produced by retokenization compare with conventional sampling? Can this diversity be leveraged for post-training to explore regions of latent space that would be otherwise inaccessible? Is there a similar phenomenon with tokenized multi-modal models? We believe this work opens up many avenues for further research.

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

## A  EXPERIMENTAL DETAILS

### A.1  STOCHASTIC RETOKENIZATION

To generate a non-canonical tokenization $E^\mu(s)$ of a string $s$, we first generate the canonical tokenization $E^\mu(s)$ using the tokenizer. Each resulting canonical token is then independently reconsidered for re-segmentation with a rate $p_{retok}$. Re-segmenting involves finding a random segmentation of a token using shorter tokens that must exist in the tokenizer vocabulary. To do this, we use an existing algorithm (Zheng et al., 2025) that samples uniformly from all valid ways of segmenting the original token. This is achieved by treating token segmentation as a recursive decomposition problem and sampling paths in the segmentation tree with probabilities proportional to the number of valid completions, ensuring that each valid segmentation is equally likely. Tokens not selected for re-segmentation ($1 - p_{retok}$) are left unchanged. The final token sequence is formed by concatenating the (possibly re-segmented) tokens in order. By tuning $p_{retok}$, we smoothly interpolate between fully deterministic canonical tokenization ($p_{retok}$=0) and fully stochastic tokenization applied to every token ($p_{retok}$=1).

## A.2 CALCULATING PASS@ METRICS

Each question $i$ from a dataset of size $N$ is replicated r=30 times for a fixed $p_{retok}$, and stochastic tokenization is applied independently to each replicate by re-segmenting canonical tokens with probability $p_{retok}$. For $p_{retok} = 0$, this replication is skipped (R=1) since it generates the canonical tokenization. We sweep over 6 values of $p_{retok} \in \{0.0, 0.2, 0.4, 0.6, 0.8, 1.0\}$, obtaining $R = (r \times 5 + 1)$ tokenizations for each prompt. The resulting tokenized inputs are then passed to the model, and inference is performed using greedy decoding (i.e., no output sampling). Thus, for each question $i$, we record how many generations (out of R) produce the correct answer. This defines pass@retok, directly analogous to pass@k, except that diversity arises from stochastic tokenization rather than output sampling. All experiments are run with the same model, prompt formatting (specific to the benchmark dataset), and decoding parameters, differing only in the retokenization probability. We report pass@retok as a function of the number of re-tokenized trials.

To calculate pass@k, we use top-p sampling ($p = 0.9, temperature = 1$) to generate R generations per question $i$.

For the figures in the paper, we plot the unbiased estimator for pass@k from (Chen et al., 2021):

$$\text{pass@k}_{emp} = \mathbb{E}_{dataset} \left[ 1 - \frac{\binom{n-c}{k}}{\binom{n}{k}} \right], \tag{1}$$

where $n$ is the total number of samples collected, and $c$ is the number of these which produce correct solutions. In some cases, the fitted $A$ is higher than the naive estimate $c/n$, which suggests exploring a larger sample set $n$. In the few cases in which we find that, sampling more cases does indeed turn out to increase the empirical value $c/n$.

For computing pass@retok, we follow almost identical logic. In this case, $n$ is the number of retokenizations, including the canonical tokenization, and $c$ is the total number of these which, under greedy decoding, generate the correct answer.

We use a dataset specific prompt formatting and answer parsing, as described in sections below.

## A.3 KNOWLEDGE RETRIEVAL

Massive Multitask Language Understanding (MMLU, introduced in (Hendrycks et al., 2021)) is a standard evaluation benchmark comprising of multiple choice questions from a wide variety of subjects. In our experiments we use a subset of 1000 questions sampled randomly from the MMLU dataset. We format each question as a zero-shot multiple-choice prompt, with the model required to output one of the possible answer choices $A, B, C, D$. Given an input like this,

```
<im_start>system
You are a helpful assistant.  For the following question, return
the answer only, without any additional reasoning or explanation.
<im_end>
<im_start>user
Question:  (ℤ, *) is a group with a * b = a + b + 1 for all a, b ∈ ℤ.  The
inverse of a is
A. 0
B. -2
C. a - 2
D. (2 + a) * -1
Answer:<im_end>
<im_start>assistant
```

we capture the first token generated by the model. If the generated token is not in $\{A, B, C, D\}$, we consider the output to be incorrect, otherwise we compare the generated answer with the true answer option. To calculate pass@retok, we only re-tokenize the tokens that represent question and its option in the above prompt, leaving the system prompt and special tokens (such as <im_start>user) in their canonical form.

### A.4 MATHEMATICAL REASONING

GSM8K (Cobbe et al., 2021) is a standard benchmark for evaluating mathematical reasoning on grade-school–level word problems. It consists of free-response questions that require computing a numerical answer rather than selecting from predefined options. In our experiments, we use a randomly sampled subset of 1000 questions from the GSM8K dataset. We do not apply any chat template to the questions as done in the previous datasets.

```
<pad><im_start>system
You are a helpful assistant.  <im_end>
<im_start>user
Question:
Given a 7-day week, how much does Alex charge for 2 weeks of
tutoring if she charges $12 per day?
Answer:
<im_end>
<im_start>assistant
```

Given this input, we generate a single completion per prompt and extract the model's predicted numeric answer from the generated text using a regular expression that selects the final number produced. The prediction is considered correct if this value exactly matches the ground-truth answer.

### A.5 CODE GENERATION

HumanEval (Chen et al., 2021) is a standard benchmark for evaluating code-generation and functional correctness on small programming tasks. It consists of 164 Python problems, each with a function signature, a natural-language docstring, and hidden unit tests. In our experiments, we evaluate on the full set of 164 problems.

```
def generate_integers(a, b):
    """
    Given two positive integers a and b, return the even digits between a
    and b, in ascending order.

    For example:
    generate_integers(2, 8) => [2, 4, 6, 8]
    generate_integers(8, 2) => [2, 4, 6, 8]
    generate_integers(10, 14) => []
    """
```

Given each prompt, we generate a single completion and extract the code for the target function. The completion is considered correct if the generated function passes all provided unit tests (i.e., functional correctness at pass@1).

#### A.5.1 CLASSIFYING COMPLETIONS

Working with the subset of retokenizations that generate passing solutions where the canonical tokenization failed, we attempt to use the retokenization to generate more passing retokenizations. To accomplish this, we begin with the initial retokenization and iteratively apply one BPE merge of tokens from the tokenizer's merge list (thus ensuring the generated tokens are present in the vocabulary). We then prompt the model with the new retokenization on the same problem and see if the retokenization generates a passing solution using the HumanEval unit test suites (Chen et al., 2021). If the solution passes the test, we accept the merge, generating a passing retokenization, and apply the next merge in the merge list. If the solution fails, we reject the merge, returning to the retokenization before the merge, and attempt to apply the next merge in the merge list. The process stops once we reach the canonical tokenization or we run out of valid merges for the retokenizations. No problems in our dataset ever reached the canonical retokenization. For problems with many initial retokenizations that generated passing solutions (HumanEval problems 26 and 159), we perform this process for each initial retokenization individually.

Python's `ast` module (Python Software Foundation, 2024) then parses each completion into an Abstract Syntax Tree (AST), with unparseable completions (syntax errors) separated from parseable completions that fail tests (logic errors) as ASTs are not able to be generated for unparseable completions (Python Software Foundation, 2024). We convert ASTs to ordered labeled trees using depth-first traversal and compute pairwise Zhang-Shasha tree edit distances (Zhang & Shasha, 1989). This algorithm calculates the minimum edit operations (node insertion, deletion, or label substitution with unit cost) required to transform one tree into another.

We compute these distances with two classification schemes. *Structural* labeling normalizes all identifier and literal tokens: variables become `Name`, constants become `Constant_int` or `Constant_str`, function definitions become `FunctionDef`, etc. *Lexical* labeling preserves token content: variables retain names (`Name:x`, `Name:y`), constants retain values (`Constant:42`), functions retain identifiers (`FunctionDef:solve`). Both schemes preserve identical tree topology; only node labels differ. This yields two $N \times N$ distance matrices quantifying pairwise tree similarity under each labeling. Across problems, we observe varying degrees of structural convergence: HumanEval/115 (52 passing solutions) collapsed into 3 distinct algorithmic patterns with 44 solutions sharing identical structure, while HumanEval/26 (13 passing solutions) showed solutions distributed across more structural categories.

| Problem | Pass | Fail | Unparsable | P-Struct | P-Lex | PF-Struct | PF-Lex |
|---|---|---|---|---|---|---|---|
| HumanEval/26 | 386 | 82 | 525 | 3 | 5, 1, 1 | 1 | 1 |
| HumanEval/72 | 0 | 0 | 75 | 1 | 1 | - | - |
| HumanEval/84 | 34 | 0 | 67 | 1 | 1 | - | - |
| HumanEval/115 | 51 | 43 | 528 | 3 | 1, 3, 1 | 9 | 1, 1, 1, 1, 2, 1, 1, 1, 1 |
| HumanEval/159 | 0 | 225 | 271 | 3 | 2, 1, 1 | 2 | 2, 1 |

Table 1: Summary of retokenization results on HumanEval problems which are not solvable by canonical plus temperature sampling. **Pass**: Total successful attempts including initial retokenization. **Fail**: Parsable failing attempts. **Unparsable**: Unparsable completions (syntax errors). **P-Struct/P-Lex**: Passing structural and lexical categories. **PF-Struct/PF-Lex**: Parsable failing structural and lexical categories. Lexical categories show comma-separated counts of lexical categories per structural category. We show data here only for tokenizations sampled by local merges starting from the passing retokenized prompts

### A.6 Visualizing Solutions to HumanEval

We construct feature vectors $\phi : \text{AST} \to \mathbb{R}^d$ as follows. For each dataset, we extract all unique AST node types appearing in all solutions. For each AST, we compute the normalized frequency of each node type, yielding a node-type distribution vector. We append four additional features: tree depth, total node count, number of unique statement types, and number of unique expression types. All features are normalized by the maximum value observed in the dataset (e.g., tree depth divided by maximum depth across all ASTs).

We apply t-SNE (van der Maaten & Hinton, 2008) directly to these $d$-dimensional vectors. Our parameters are: perplexity $= \min(30, N - 1)$ where $N$ is the number of solutions, maximum iterations $= 1000$, which caps the number of adjustment steps, and we fixed our random seed $= 42$. The resulting 2D embeddings are plotted with points colored by test outcome, passing or wrong logic failing. Spatial clustering patterns vary by problem: HumanEval/115 shows tight clustering of passing solutions with clear separation from failures, while HumanEval/72 and HumanEval/84 show passing solutions more dispersed in the embedding space with less distinct cluster boundaries.

## B Combinatorics of Tokenization

We provide some more details of tokenization for illustrative purposes. The first interesting observation is that every token in the token vocabulary has a number of retokenizations that grows exponentially in the string length of the token (see Fig.4(Left)). For a prompt with $N$ canonical tokens, which have on average about $n$ characters per token, the space of tokenizations will scale as $O(e^{Nn})$, which is enormous. We are therefore sampling only a very small subset of these tok-

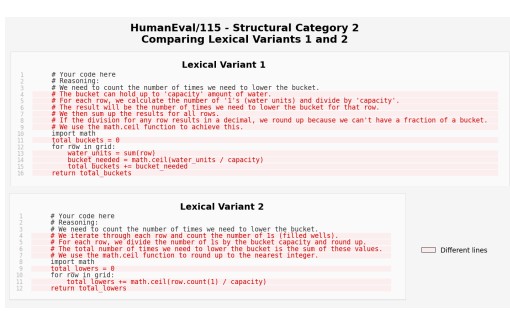

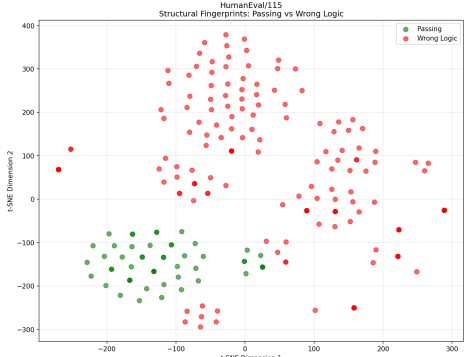

Figure 3: Hierarchical clustering reveals that 52 passing solutions collapse into 3 structural categories under AST comparison, with the dominant category containing 44 solutions that share identical algorithmic structure. (**Left**) Stacked comparison of two lexical variants from Structural Category 2 of HumanEval problem 115 shows differences in surface-level implementation details while the underlying algorithmic logic remains identical. (**Right**) t-SNE projection of AST structural features shows passing solutions (green) forming a tight cluster while wrong logic solutions (red) are dispersed across the embedding space, indicating that successful retokenizations converge on similar algorithmic structures. The clear spatial separation demonstrates that structural features predict correctness, though a few failures near the passing cluster suggest subtle logical errors rather than fundamental structural differences.

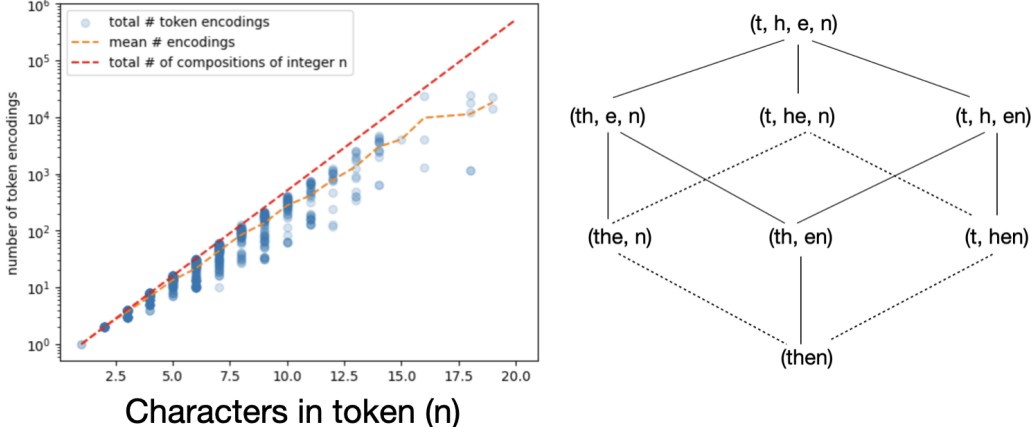

Figure 4: (Left) Each individual token of length $n$ in the token vocabulary has a number of retokenizations that grows exponentially with $n$. The upper bound (dashed red) is set by the total number of partitions of a string of length n, which is $2^{n-1}$.Right) A diagram showing all possible tokenizations of the surface string "then". Solid lines connect tokenizations which can be connected by a single merge that exists in the BPE merge list. Dashed lines connect tokenizations which cannot be connected by a merge from the tokenizer merge list. These data are gathered from the OLMo 2 tokenizer.

enizations. The behavior we observe must be understood as typical behavior, rather than average behavior. In Fig.4(Right), we show all the retokenizations of the single token "then", which happens to have the maximum number of possible tokenizations. This is because all of the possible subtokens are also elements of the token vocabulary. The solid lines connect tokenizations that differ by the application of a single merge from the merge-list. Dashed lines connect tokenizations which cannot be connected by any single existing merge.

## C   Tokenization Invariance Implies Byte-Level generation

Here we argue that a model which is required to be invariant to tokenization must ultimately generate text at the byte-level. A language model consists of a parameterized probability function $P_\theta(s)$ which assigns a probability to a string. Tokenized language models assign the probability to a token sequence $E^0(s)$ instead of directly to the byte-string. Therefore, the argument to the parameterized probability is a sequence of integers, which is the token encoding of $s$: $P_\theta(E^0(s))$. Requiring invariance to tokenization means requiring that this probability is independent of how the byte-string is encoded, i.e. there exists a byte-string probabiliy $P(s)$ such that

$$P_\theta(E^\mu(s)) \equiv P(s) \quad \forall \mu \in \{0, ..., L(s) - 1\} \tag{2}$$

Given this, let us consider generating text from this distribution. Given a context $s$, we are interested in the conditional probability of the next token $P_\theta(w|s) = P_\theta(w, s)/P_\theta(s)$. We ask what is the next token that is likely to be generated. Tokenization invariance implies that $P_\theta(w, s)$ is invariant to tokenization. Then consider two different tokenizations of $w$: $E^0(w) = (w_1)$, and $E^1(w) = (x_1, x_2)$, then

$$P_\theta(E^0(w), E(s)) = P_\theta(w_1|s)P_\theta(s) \tag{3}$$

$$P_\theta(E^1(w), E(s)) = P_\theta(x_2|x_1; s)P_\theta(x_1|s)P_\theta(s) \tag{4}$$

Tokenization invariance implies $P_\theta(E^0(w), E(s)) = P_\theta(E^1(w), E(s))$. Since conditional probabilities must be smaller than one, this then implies

$$P_\theta(x_1|s) > P_\theta(w_1|s) \tag{5}$$

Thus the subtoken $x_1$ has a higher probability than the composite token $w_1$. If $x_1$ can be further divided into subtokens, then we can repeat this analysis and find that the smaller subtoken will always have higher probability. Since every token can be broken into single characters, this implies that the most probable next token will always be a single-character. Therefore, in greedy generative mode, such an explicitly tokenization invariant model will generate only byte-level tokens.

