# OpenReview forum: "Generating output diversity from prompt re-tokenization"
_ICLR.cc/2026/Workshop/Sci4DL — Sci4DL 2026_

### Official Review · Reviewer_fk9i · 2026-02-26

**Fit:** 1
**Significance:** 1
**Confidence:** 2

**Summary:**

This paper introduces *tokenization sampling*: a test-time strategy that leverages the fact that a given string might admit multiple valid segmentations under the same tokenizer. The novel sampling technique proposed by the authors retokenizes the input prompt stochastically, while using greedy decoding for generation. The authors define an evaluation metric, pass@retok, analogous to pass@k, where diversity of the output arises from different input tokenizations rather than independent output sampling.

Experiments show that the average pass@retok increases with k, suggesting that the proposed sampling procedure *can* generate diverse outputs. While the average pass@retok accuracy is consistently lower than pass@k; there is a subset of problems (especially for the knowledge retrieval benchmark MMLU) where retokenization is better than output resampling. The paper frames these findings as evidence of an emergent approximate “tokenization symmetry” in trained LLMs and positions retokenization as a form of test-time data augmentation.

**Strengths:**

**Clean idea.** The proposal to treat alternative tokenizations as a source of test-time diversity is simple, well-motivated, and easy to implement without modifying model weights.

**Non-trivial empirical observation.** The finding that different tokenizations of the same surface string can lead to different greedy decoding trajectories is interesting. In particular, the fact that some problems are solved only under retokenization suggests that models are not invariant to token boundaries in practice.

**Clear experimental protocol.** The paper defines pass@retok carefully and mirrors the standard pass@k estimator, making comparisons transparent. The evaluation spans multiple domains (knowledge retrieval, math reasoning, and code generation), which strengthens the empirical case.

**Suggestions:**

**Clarify difference with other input perturbations.** While the empirical results are suggestive, it remains unclear how *tokenization sampling* differs in effect from other forms of input perturbation or prompt augmentation. A comparison against simple baselines (e.g. paraphrasing, or higher temperature sampling under equal compute) would help clarify whether retokenization provides unique benefits.

**Clarify when retokenization helps.** The results show that only a subset of problems benefits from the new sampling method. A deeper analysis of these cases (e.g., error types and prompt length) would significantly strengthen the contribution and help explain why retokenization unlocks solutions in some instances but not others.

**Clarify Tokenization Symmetry.** The paper repeatedly interprets the results as evidence for an emergent tokenization symmetry (e.g., Sections 1 and 4). However, Appendix C argues that strict tokenization invariance would imply that generation collapses to byte-level tokens under greedy decoding. If invariance held exactly, different encodings should produce identical conditional distributions and therefore identical greedy outputs. Yet Figures 1C–1D show that retokenization frequently changes outcomes, and in some cases enables solutions that canonical tokenization cannot produce. This indicates that the model is not invariant to re-tokenization. In fact, the effectiveness of retokenization appears to rely precisely on residual sensitivity to token boundaries. This tension needs to be reconciled.

**Clarify workshop alignment.** The workshop call emphasizes the scientific method: forming hypotheses about deep learning and designing controlled experiments to validate or falsify them. While this paper touches on a potentially interesting hypothesis (tokenization invariance), it does not clearly articulate it or rigorously test it. Instead, the bulk of the empirical contribution in Section 3 evaluates tokenization sampling as a test-time technique via pass@retok comparisons. Because pass@k consistently achieves higher asymptotic performance (Figure 1C), the method does not appear to constitute a superior alternative to standard output sampling. Without deeper analysis of representational behavior or formal tests of invariance with respect to re-tonenization, the work reads primarily as a proof-of-concept demonstration of a new test-time technique, which does not fit within the scope of this workshop.

---

### Official Review · Reviewer_UpmZ · 2026-02-27

**Fit:** 3
**Significance:** 2
**Confidence:** 3

**Summary:**

This paper studies non-deterministic tokenization as a source of generation diversity. Given a prompt, it proposes tokenization sampling, which retokenizes the prompt into different token sequences and then generates responses using greedy decoding. The work evaluates pass@retok(k) across multiple tasks and checkpoints from different training stages. Results show that tokenization sampling yields diverse outputs, as pass@retok(k) increases with larger k, analogous to pass@k. Moreover, tokenization sampling enables LLMs to solve some problems that conventional temperature sampling fails to solve even with large k.

**Strengths:**

Treating non-deterministic tokenization as a mechanism for generation diversity is novel to my knowledge. Although the current empirical results suggest that retokenization can degrade performance on some tasks, the idea opens substantial room for further investigation. The evaluation setup is also clear and reasonably comprehensive.

**Suggestions:**

1. It would be valuable to analyze more closely which types of retokenization lead to correct solutions that temperature sampling does not find, and which types make problems harder.

2. It may be interesting to compare different retokenization strategies, and to consider applying retokenization during generation (for example, retokenizing the partial output and continuing decoding).

---

### Official Review · Reviewer_DmZD · 2026-02-27

**Fit:** 2
**Significance:** 2
**Confidence:** 3

**Summary:**

I want to slightly reframe what I think this paper is doing, because I do not think the current framing "correctly" captures it (in my mind). From my perspective, this work is really about trajectory diversity, specifically internal trajectory diversity that arises from choosing different subword segmentations within a fixed vocabulary.

I am specifically choosing those words; because the current title implies something broader than what is being explored.

The phrase “prompt retokenization” is broad enough that it can easily be interpreted as adding new tokens, adapting the vocabulary, or constructing specialized tokens. That is not what is happening here. The model’s vocabulary is fixed. No new tokens are introduced. What changes is the segmentation of the same byte-level string into different combinations of existing subwords. I think making this distinction clearer earlier would help position the contribution more precisely.

In the LLM literature, when we talk about diversity, we usually mean content diversity. For example, if someone asks for a story, we might want different stories each time. Temperature is one mechanism for that, and there are reward-based and decoding-based approaches as well. What this paper focuses on is something different. There is usually still one correct output. The diversity here is not creative variation, but diversity in the internal search process. By sampling over tokenizations rather than output distributions, the authors increase the number of internal trajectories explored. Larger k increases the probability that one of those trajectories lands in the correct basin, similar in spirit to pass@k.

For that reason, I am not entirely convinced by the phrase “output diversity.” What I see empirically is segmentation-dependent sensitivity. Some easy problems become slightly harder, and some previously unsolved problems become solvable. That looks more like increased sensitivity to how subwords are formed than invariance to tokenization.

This is also why I am somewhat confused by the invariance discussion. The authors show that a tokenization-invariant model is equivalent to a byte-level model. I understand the argument, and it is mathematically correct. But the empirical gains here seem to rely on segmentation sensitivity, not invariance. I am unclear why the invariance argument is central to this paper, or what conceptual role it is meant to play beyond being an interesting observation.

Overall, I think the idea is interesting and novel, but I believe the framing, related work positioning, and empirical analysis could be sharpened.

**Strengths:**

1.	Interesting and clean core idea
Treating tokenization as a sampling axis instead of treating temperature as the only source of stochasticity is conceptually interesting. The pass@retok formulation is clean and parallels pass@k in an intuitive way.
	2.	Clear empirical setup across domains
The evaluation spans MMLU, GSM8K, and HumanEval, and the computation of pass@retok is well described.
	3.	Evidence of segmentation-dependent effects
The observation that some problems are solvable only under alternative tokenizations is compelling and supports the trajectory-based interpretation.
	4.	Training-stage analysis
The comparison across Base, SFT, DPO, and Instruct variants is valuable. The emergence of stronger tokenization symmetry after post-training is an interesting empirical result.
	5.	Structural analysis in the appendix
The AST-based clustering analysis for code solutions is thoughtful and adds depth to the empirical findings.

**Suggestions:**

1.	Invariance discussion feels disconnected from the empirical mechanism
While the mathematical argument about tokenization invariance implying byte-level modeling is correct, I am not convinced that it is central to the paper’s contribution. The gains demonstrated seem to arise from sensitivity to segmentation, not invariance. Clarifying this relationship would strengthen the conceptual coherence of the paper.
	2.	Over-explained formalism and under-explained practical insights
The notation and nomenclature section is technically correct, but it feels somewhat over-elaborated relative to its importance. At the same time, there is limited analysis of which tokenization patterns are actually beneficial. A more focused empirical breakdown of segmentation families would be more impactful than extended formalism.
	3.	Missing related work in tokenization interventions
I believe the related work section omits several relevant papers, including AdaptBPE, Fusion Token, dynamic tokenization approaches, zip2zip, and especially Broken Tokens. While many of those works focus on efficiency or fairness rather than search diversity, they still intervene on tokenization at inference time. Broken Tokens in particular studies performance shifts under non-canonical segmentations, which feels closely related. A more explicit comparison would strengthen positioning.
	4.	Large k values reduce practical relevance
pass@k is typically reported for small values such as 1, 5, or 10. In this paper, k is swept up to 140. It is difficult to see how running 140 inference passes per question is practical. While the curves are informative analytically, the practical implications would be stronger if improvements were meaningful at small k, or if there were a principled way to select promising tokenizations.
	5.	Benchmark choice and saturation
HumanEval, GSM8K, and MMLU are widely used and relatively mature benchmarks. Given the models evaluated, it would be useful to see results on more challenging or contemporary benchmarks. The current evaluation does not fully demonstrate the claim of unlocking otherwise inaccessible regions of behavior.
	6.	Input tokenization analysis is limited
The appendix includes structural clustering of output ASTs, which is interesting. However, I may have partially misunderstood Figure 3, because I understood it as focusing on solution clustering rather than tokenization clustering. If the authors have already performed input-level grouping analysis, it is not clearly highlighted. It would be valuable to know whether successful tokenizations share identifiable structural properties at the segmentation level.
	7.	Post-training effects are under-explained
The improvement of pass@retok across Base, SFT, DPO, and Instruct variants is shown, but the intuition behind why post-training amplifies segmentation sensitivity is not deeply explored. More discussion here would strengthen the narrative.
	8.	Limited comparison to other diversity methods
The method is primarily compared to temperature sampling (which makes sense). It would be helpful to compare this approach to other diversity-goal techniques, and maybe look at those in scenarios where multiple correct outputs are possible. This would clarify whether the gains are specific to subwords methods proposed here or are reflective of broader search augmentation effects.

---

### Meta-Review · Area_Chair_6rY7 · 2026-03-01

**Recommendation:** Accept

**Metareview:**

This work is well motivated, has interesting empirical observations and a strong empirical methodology. It however suffers from lack of clarity so I suggest the authors review the feedback from Reviewer fk9i and update their manuscript for the camera-ready version to include the requested clarifications. I recommend acceptance.

---

### Decision · Program_Chairs · 2026-03-02

Accept